# Sexual behaviour, changes in sexual behaviour and associated factors among women at high risk of HIV participating in feasibility studies for prevention trials in Tanzania

**Diana Faini** [1,2,3]*, **Claudia Hanson**[2,4], **Kathy Baisley**[3], **Saidi Kapiga**[3,5], **Richard Hayes**[3]

**1** Department of Epidemiology and Biostatistics, Muhimbili University of Health and Allied Sciences (MUHAS), Dar es Salaam, Tanzania, **2** Department of Global Public Health, Global Health—Health Systems and Policy Research, Karolinska Institutet, Stockholm, Sweden, **3** Department of Infectious Disease Epidemiology, London School of Hygiene and Tropical Medicine, London, United Kingdom, **4** Department of Disease Control, London School of Hygiene and Tropical Medicine, London, United Kingdom, **5** Mwanza Intervention Trials Unit, Mwanza, Tanzania

* fainidiana@gmail.com

**Data Availability Statement:** Data cannot be shared publicly due to ethical restrictions as it contains identifying and sensitive information.

## Abstract

### Introduction

Risk reduction towards safer behaviour is promoted after enrolment in HIV prevention trials. We evaluated sexual behaviour, changes in sexual behaviour and factors associated with risky behaviour after one-year of follow-up among women enrolled in HIV prevention trials in Northern Tanzania.

### Methods

Self-reported information from 1378 HIV-negative women aged 18–44 enrolled in microbicide and vaccine feasibility studies between 2008–2010,was used to assess changes in behaviour during a 12-month follow-up period. Logistic regression with random intercepts was used to estimate odds ratios for trends in each behaviour over time. A behavioural risk score was derived from coefficients of three behavioural variables in a Poisson regression model for HIV incidence and thereafter, dichotomized to risky vs less-risky behaviour. Logistic regression was then used to identify factors associated with risky behaviour at 12 months.

### Results

At baseline, 22% reported multiple partners, 28% were involved in transactional sex and only 22% consistently used condoms with non-regular partners. The proportion of women reporting multiple partners, transactional sex and high-risk sex practices reduced at each 3-monthly visit (33%, 43% and 47% reduction in odds per visit respectively, p for linear trend <0.001 for all), however, there was no evidence of a change in the proportion of women

These restrictions have been imposed by the ethics committees of the National Institute for Medical Research,Tanzania. Data are available on request from the MITU Data Access Committee (contact via info@mitu.or.tz) for researchers who meet the criteria for access to de-identified data.

**Funding:** The microbicides-preparedness study was funded by the European and Developing Countries Clinical Trials Partnership (EDCTP; project code: CT_ct_05_32070_002) and the vaccines-preparedness study was funded by EDCTP and the Bill and Melinda Gates Foundation (project code: CG_ct_06_33111). The funders had no role in study design, data collection and analysis, decision to publish, or preparation of the manuscript.

**Competing interests:** The authors have declared that no competing interests exist.

consistently using condoms with non-regular partners (p = 0.22). Having riskier behaviours at baseline, being younger than 16 years at sexual debut, having multiple partners, selling sex and excessive alcohol intake at baseline were strongly associated with increased odds of risky sexual behaviour after 12 months (p<0.005 for all).

## Conclusion

An overall reduction in risky behaviours over time was observed in HIV prevention cohorts. Risk reduction counselling was associated with decreased risk behaviour but was insufficient to change behaviours of all those at highest risk. Biological HIV prevention interventions such as PrEP for individuals at highest risk, should complement risk reduction counselling so as to minimize HIV acquisition risk.

## Introduction

Women with high risk behaviour and at high risk of HIV acquisition are often recruited to participate in HIV prevention trials[1,2], in part to ensure that the required sample size or follow-up time for adequate study power are not too large[3]. In such trials, researchers are obliged, for ethical reasons, to provide HIV risk reduction services such as free condoms, treatment of sexually transmitted infections (STIs) and risk reduction counselling, to minimize the risk of HIV acquisition [4–6]. However, experience has shown [7–12] that, while risk behaviour decreases in many participants during these trials, this does not apply to all participants. Understanding the characteristics of women who continue to engage in risky behaviour (or even increase their risk behaviour) would be useful so that intensified risk reduction counselling can be given to these participants, and special efforts could be made to ensure optimal follow-up of those participant as they are more likely to acquire HIV.

Studies among participants in HIV vaccine trials have documented that participants may continue to engage in risky behaviour due to optimism about high efficacy of the intervention for HIV prevention (termed 'risk compensation') [10, 13, 14]. Unfortunately, it is not possible to assess the effect of trial participation itself on changes in risk behaviour, because most HIV prevention trials do not include comparison groups who receive no trial product (either active or placebo). Several longitudinal cohorts have described risky behaviour profiles of participants at enrolment, however trends in sexual behaviour during follow-up are under-explored.

We therefore used data from cohorts of HIV-negative women participating in feasibility studies for HIV prevention trials in Northern Tanzania to describe risky behaviour profiles of women at enrolment, evaluate patterns of their sexual behaviours over time, and determine factors associated with engaging in risky behaviours after one year. We sought to test the hypothesis that risky sexual behaviour among women participating in HIV prevention trials decreases over time. The epidemiology of HIV and STIs among women in these cohorts has been described previously [15].

## Methods

### Study design

We used data from two observational cohort studies conducted between 2008–2010; a vaccine feasibility study which was carried out in Moshi town and a microbicide feasibility study in the Northern lake zone towns of Geita, Shinyanga, and Kahama. Both studies used a similar design

and data collection instruments. HIV negative women aged 18–44 years working in bars, restaurants, guesthouses, hotels, shops selling traditionally brewed beer and food-sellers (popularly known as *mamalishe)* were recruited. Some women in these occupational groups are reported to periodically supplement their income through transactional sex, although are not necessarily perceived as commercial sex workers within the wider community. The HIV and HSV-2 incidence among women in these studies were 3.7/100 person-years and 28.6/100 person-years over 12 months [15].

Women were invited to enrol through a study clinic located within their towns. Upon their visit, women were informed about the study, screened for eligibility, and offered HIV testing with pre and post-test counselling. Women were eligible for enrolment if they were aged 18–44 years, willing to undergo HIV testing and to receive results and not planning to move away for the duration of the study. Eligible and consenting women underwent face- to-face interviews where information on their socio-demographic characteristics, employment, reproductive history and work mobility was collected. Information about alcohol use was obtained using the Alcohol Use Disorders Identification Test (AUDIT) [16]. Further details of the study population, recruitment and procedures have been reported previously [15, 17, 18]. In brief, all women underwent physical and genital examination at enrolment and follow-up clinical visits. Blood and genital samples were collected and tested according to standard operating procedures at the study laboratories. Women with symptoms and clinical signs of STIs were managed according to Tanzanian national syndromic management guidelines[19]. HIV voluntary counselling and testing, risk reduction counselling, treatment for medical problems and condoms were provided during each scheduled follow-up visit.

In both cohorts, HIV rapid testing was performed in parallel using SD Bioline HIV-1/2 3.0 (Standard Diagnostics, Inc., Korea) and Determine HIV-1/2 (Alere Medical, Co., Ltd, Japan) tests. If the rapid tests were positive or discordant, HIV infection was confirmed in the respective laboratories using either third generation Murex HIV 1.2.O (Abbott UK, Dartford, Kent, England) or Vironostika HIV Uniform II plus O(bioMérieux Bv, The Netherlands) ELISAs. Participants were followed every three months for 12 months with similar interviews, clinical examination, and collection of samples performed at each visit.

## Statistical analysis

Socio-economic and behavioural risk factors were categorised based on their distributions to minimize data sparsity. AUDIT responses from 10 questions were dichotomised as score <8 points (non-drinker or low-risk drinking) versus ≥8 points (harmful or hazardous drinking).

Frequency of condom use with regular partners and/or non-regular partners was categorized as (1) "consistent" (2) "inconsistent" and (3) "never" if women reported (1) *always*, (2) *sometimes* or *often use* or (3) not using condoms, respectively.

A composite variable termed 'high-risk sex' was generated combining women reporting a history of rape, sex during menses or anal sex in the past three months. Chi-squared tests were used to compare the distribution of socio-demographic characteristics and sexual behaviours at baseline between younger (aged <25 years) and older (≥25years) women.

Prevalence of self-reported sexual behaviours at the baseline, 3, 6, 9 and 12-month visits were examined. Logistic regression with random intercepts (for women) was used to estimate odds ratios (OR) and 95% confidence intervals (CIs) for the change in the prevalence of each behaviour over time. The random effect models were used so as to account for correlations between repeated measurements at the multiple visits. Women who missed visits contributed data to the analysis for all visits at which they were seen. Study visit was included in the model

as a linear term (coded as 1,2,3,4,5) to obtain ORs for the linear trend in behaviour across study visits.

A score for risky sexual behaviour was developed using regression coefficients from a Poisson regression model with HIV seroconversion as outcome. In this analysis, the date of seroconversion was assumed to take place midway between the last negative and first positive HIV serology results. Women were censored at the earliest date of HIV seroconversion, date last seen, or end of follow up. Baseline sexual behaviour factors whose univariate association reached statistical significance at $p<0.05$ were entered into the multivariable model. Variables that remained statistically significant in this model were considered for inclusion in the risk score. All multivariable models were adjusted for age, study area, and marital status as a priori confounders.

Selection of the "best" multivariable model was based on one that provided negative regression coefficients for behaviours known to be protective and positive regression coefficients for less safe behaviours. For instance, a model that had a negative coefficient for transactional sex, i.e. indicating transactional sex was protective against HIV acquisition, was considered implausible. This selective approach to model building was chosen for two main reasons: (i) so as to generate predicted values of sexual behaviour risk scores in their appropriate category (protective vs harmful); (ii) with such a small number of HIV seroconversion events (44 HIV cases), stringent criteria were needed to select few "key" parameters for the model.

A risk score was then generated for each woman at baseline and at 12 months, based on the regression coefficients for three baseline sexual behaviours (number of partners in the last three months, condom use at last sex and high-risk sex) in the final model. Thereafter, distribution of the risk scores was examined at baseline and 12 months to determine the mean risk score at each visit. Using the mean of the baseline risk score (i.e. -5.05) as a cut-off point, the baseline and 12-month behaviour risk scores were dichotomised into "risky" sexual behaviour (risk score values $>$ -5.05) and "less-risky" sexual behaviour (risk score values $\leq$ -5.05). Lastly, logistic regression was used to examine each covariate and its association with having risky behaviour at 12 months. Because few women were categorised as having risky behaviour at 12 months, no attempt was made to build a full multivariable model, to avoid problems with data sparsity. Each covariate was therefore only adjusted for baseline behaviour.

## Ethical considerations

Ethical clearance for this analysis was obtained from the London School of Hygiene & Tropical Medicine (LSHTM MSc Ethics Ref: 12155). Ethical approvals for the cohort studies were granted by the Ethics Committees of the National Institute for Medical Research, Kilimanjaro Christian Medical Centre and LSHTM (Approval number 5439 and 5188). Informed consent (signature or witnessed thumbprint) was obtained from all eligible participants after careful explanation of the study aims and procedures. Free reproductive health services, including syndromic management of STIs, family planning, health education, and voluntary HIV counselling and testing were provided.

## Results

### Sociodemographic and sexual behavioural characteristics at baseline

Of the 1,378 HIV negative women in the combined cohort, 966 (70%) were enrolled in the microbicide feasibility study in Northern lake zone (375 in Geita, 306 Kahama and 285 in Shinyanga) and 412 (30%) were enrolled in the vaccine feasibility study in Moshi. Most of the women in the cohort were unmarried (74%) and 26% had been sexually active before the age of 16 (Table 1). Over a quarter of women reported having offered sex in return for a gift or

**Table 1. Baseline socio demographic and sexual behaviour characteristics of women enrolled in HIV prevention trials in Northern Tanzania, and comparison between younger (<25 years) and older (≥25 years) women.**

| | All | Age <25years | Age ≥25years | p-value |
|---|---|---|---|---|
| | N(Column %) | N(Column %) | N(Column %) | $(x^2)$ |
| **Overall** | 1378 | 536 | 842 | |
| **Study cohort** | | | | |
| Microbicide feasibility study | 966(70) | 392(73) | 574(68) | 0.05 |
| Vaccine feasibility study | 412(30) | 144(27) | 268(32) | |
| **Education level** [†] | | | | |
| None/Primary | 1149(83) | 422(79) | 727(86) | <0.001 |
| Secondary/Tertiary | 227(16) | 112(21) | 115(14) | |
| **Marital Status** | | | | |
| Married | 357(26) | 68(13) | 289(34) | |
| Separated/divorced/widowed | 580(42) | 160(30) | 420(50) | <0.001 |
| Single | 441(32) | 308(57) | 133(16) | |
| **Age at sexual debut(years)** | | | | |
| <16 | 362(26) | 167(31) | 195(23) | |
| ≥16 | 916(67) | 334(62) | 582(69) | 0.004 |
| Missing | 100(7) | 35(6) | 65(8) | |
| **Number of lifetime partners** [†] | | | | |
| 0–4 | 715(52) | 282(53) | 433(51) | |
| 5–9 | 261(19) | 100(19) | 161(19) | 0.55 |
| ≥10 | 189(14) | 68(13) | 121(14) | |
| Don't know | 209(15) | 83(16) | 126(15) | |
| **Number of partners, past 3m** [†] | | | | |
| 0 | 81(6) | 29(5) | 52(6) | |
| 1 | 982(71) | 360(67) | 622(74) | 0.02 |
| 2 | 195(14) | 86(16) | 109(13) | |
| 3+ | 104(8) | 53(10) | 51(6) | |
| **Condom use at last sex** [†] | | | | |
| Yes | 630(46) | 307(57) | 323(39) | <0.001 |
| No | 747(54) | 228(43) | 519(62) | |
| **Condom use (regular partners), past 3m** [†] | | | | |
| Consistently | 399(29) | 193(36) | 206(25) | |
| Inconsistently | 239(17) | 117(22) | 122(15) | <0.001 |
| Never | 566(41) | 157(29) | 409(58) | |
| Not had sex in past 3m | 161(12) | 60(11) | 101(12) | |
| **Condom use (non-regular partners),past3m** [†] | | | | |
| Consistently | 306(22) | 145(27) | 161(19) | |
| Inconsistently | 99(7) | 46(9) | 53(6) | <0.001 |
| Never | 175(13) | 58(11) | 117(14) | |
| No sex with non-regular partners | 785(57) | 278(52) | 507(60) | |
| **Transactional sex, past 3m** [†] | | | | |
| Yes | 386(28) | 185(35) | 201(24) | <0.001 |
| No | 985(72) | 346(64) | 639(76) | |
| **High-risk sex, past 3 months** [†] [‡] | | | | |
| Yes | 122(9) | 62(12) | 60(7) | 0.02 |
| No | 1254(91) | 473(88) | 781(93) | |
| **AUDIT score** [†] [§] | | | | |

*(Continued)*

**Table 1.** (Continued)

| | All | Age <25years | Age ≥25years | p-value |
|---|---|---|---|---|
| | N(Column %) | N(Column %) | N(Column %) | ($x^2$) |
| Non-drinker or low-risk | 1186(86) | 457(83) | 729(87) | 0.79 |
| Harmful or hazardous drinking | 185(13) | 76(14) | 109(13) | |
| **HIV risk perception** [†] | | | | |
| Small/No risk | 689(50) | 289(54) | 400(48) | 0.01 |
| Moderate/Great risk | 342(25) | 107(20) | 235(28) | |
| Don't know | 343(25) | 138(26) | 205(24) | |

† Missing values; Education level and High-risk sex, past 3 months = 2 missing values; Number of lifetime partners = 4 missing values; Number of partners, past 3m = 16 missing values; Condom use at last sex = 1 missing value; Condom use (non-regular partners),past 3m = 13 missing values; Transactional sex, past 3m and AUDIT score = 7 missing values

[‡ ] High-risk sex was defined as women reporting a history of rape, sex during menses or anal sex in the past three months

[§ ]AUDIT score estimated among women reporting use of alcohol in the last 12 months from. The score was estimated from 10 questions and were dichotomised as score <8 points (non-drinker or low-risk drinking) versus ≥8 points (harmful or hazardous drinking).

money (28%), and 9% reported having engaged in high risk sex in the past three months. Younger women were more educated, had an earlier age of sexual debut, reported more condom use, and were more likely to engage in transactional and high-risk sex compared to older women.

## Prospective changes in self-reported risky sexual behaviours

Attendance at the 3, 6, 9 and 12 months visit was 84%, 80%, 80% and 86% of participants, respectively (Table 2). There was strong evidence of a reduction in most reported risky sexual

**Table 2. Changes in reported sexual behaviours over time in the cohort, among women attending the study visits.**

| | Baseline(%) | 3m (%) | 6m (%) | 9m (%) | 12m (%) | OR [†] | P LRT[*] |
|---|---|---|---|---|---|---|---|
| **No. of participants(N)** | 1378(100) | 1154(84) | 1102(80) | 1107(80) | 1184(86) | - - | - - |
| **Two/more partners past 1m** | 154(13) | 90(8) | 109(10) | 104(10) | 86(7) | 0.82 (0.75–0.89) | <0.001 |
| **Two/more partners past 3m** | 299(22) | 152(13) | 137(12) | 120(11) | 111(9) | 0.67 (0.63–0.73) | <0.001 |
| **High-risk sex, past 3m** | 122(9) | 63(5) | 41(4) | 32(3) | 17(1) | 0.57 (0.50–0.64) | <0.001 |
| **Transactional sex, past 3m** | 386(28) | 183(16) | 120(11) | 94(9) | 71(6) | 0.5 (0.49–0.57) | <0.001 |
| **Condom use at last sex (Yes)** | 630(46) | 465(40) | 402(36) | 392(35) | 419(35) | 0.8 (0.78–0.87) | <0.001 |
| **Any Condom use in past 3m**[‡] | 775(60) | 588(53) | 526(50) | 491(47) | 509(47) | 0.7 (0.72–0.80) | <0.001 |
| **Condom use (regular partners) past 3m (consistently)**[§] | 399(33) | 309(31) | 242(25) | 241(25) | 226(22) | 0.77 (0.72–0.82) | <0.001 |
| **Condom use (non-regular partners), past 3m (consistently)** [ꞡ] | 306(53) | 197(57) | 154(54) | 122(52) | 131(56) | 0.95 (0.87–1.03) | 0.22 |
| **Women reporting non-regular partners in the past 3m** | 580(42) | 346(30) | 285(26) | 234(21) | 235(20) | 0.6 (0.63–0.70) | <0.001 |
| **Moderate/greater HIV risk perception** | 342(25) | 296(26) | 348(32) | 363(33) | 429(36) | 1.1 (1.20–1.33) | 0.01 |

OR = odds ratio. (%) Proportion restricted to non-missing data in respective visits therefore may not add up to total(N). High-risk sex was defined as women reporting a history of rape, sex during menses or anal sex in the past three months.

[†] OR per study visit from Random-effects logistic regression model with visit as a linear term.

[*] Likelihood ratio test of significance for a linear trend.

[‡]Analysis restricted to women reporting to have had sex in the last 3 months.

[§]Analysis restricted to women reporting having a regular sex partner in the last 3 months; proportions and OR comparing consistent condom use versus Never/inconsistent use.

[ꞡ]Analysis restricted to women reporting having a non-regular sex partner in the last 3 months; proportions and OR comparing consistent versus Never/inconsistent condom use

behaviours over time. At each visit, there was a 33% reduction in the odds of reporting two or more partners in the preceding three months (OR 0.67, 95%CI = 0.63–0.73, p for linear trend<0.001). There was also a 47% decrease in the odds of reporting transactional sex at each study visit (OR 0.53, 95%CI = 0.49–0.57, p for linear trend<0.001). However, we also observed a decrease in any condom use over time, and in consistent condom use with regular partners (OR 0.77, 95%CI = 0.72–0.82, p for trend <0.001). In contrast, there was no evidence of a change in the odds of consistent condom use with non-regular partners (OR 0.95, 95% CI = 0.87–1.03 p = 0.22). There was also evidence of an increase in the proportion of women with higher HIV risk perception over time (OR 1.13, 95%CI = 1.20–1.33 p = 0.01).

## Sexual behaviour risk scores for baseline and 12 month visits

Sexual behaviour risk scores at baseline calculated based on the coefficients obtained from the Poisson regression model (Table 3) ranged from -6.06 (less-risky behaviour) to -3.41 (more-risky behaviour) with a mean risk score of -5.05 (95%CI = -5.08 to -5.03). Mean risk scores at 12 months were slightly lower, -5.12 (95%CI = -5.14 to -5.11), indicating less-risky behaviour

**Table 3. Results of Poisson regression model of HIV incidence used to develop sexual behaviour risk score.**

| Predictor | HIV infected /person years | Rate per 100pyr (95% CI) | Coefficient (adjusted log HR) (SE) | HR(95%CI) | Adjusted HR N = 1,361 |
|---|---|---|---|---|---|
| No. of partners in the last 3m | | | | | |
| 0 | 1/69 | 1.5 (0.2–10.3) | 0 | 1 | 1 |
| 1 | 28/859 | 3.3 (2.3–4.7) | 0.89(1.03) | 2.25(0.31–15.52 | 2.43(0.32–18.15) |
| 2 | 5/161 | 3.1 (1.3–7.5) | 0.70(1.11) | 2.14(0.25–18.35) | 2.01(0.23–17.55) |
| 3+ | 10/81 | 12.3 (6.6–22.9) | 1.93(1.07) | 8.47(1.08–66.19) | 6.88(0.84–56.19) |
| Condom use at last sex | | | | | |
| No | 22/670 | 3.3(2.2–5.0) | 0 | 1 | 1 |
| Yes | 22/511 | 4.3(2.8–6.5) | -0.16(0.32) | 1.31(0.73–2.37) | 0.85(0.45–1.60) |
| High-risk sex, past 3m | | | | | |
| No | 36/1082 | 3.3 (2.4–4.6) | 0 | 1 | 1 |
| Yes | 8/98 | 8.2 (4.1–16.4) | 0.56(0.43) | 2.47(1.15–5.31) | 1.75(0.76–4.03) |
| Intercept | -- | -- | -5.90(1.19) | | |

Coefficient = Poisson regression coefficient; SE = standard error. High-risk sex was defined as women reporting a history of rape, sex during menses or anal sex in the past three months.

Model adjusted for three priori confounders i) age (< 25 vs ≥25+years), ii) town (Moshi vs Lake zone) and iii) marital status (married vs single/separated/widowed/divorced). Binary variables coded 0 for no or 1 for yes. The regression coefficient are log (rate ratio) for change of 1 unit in the corresponding variable. The sexual behaviour risk score values for baseline and at 12 months were obtained from Poisson regression model and generated using the following equation:

$$
\begin{aligned}
\textbf{\textit{Risk score}} = -5.90 &+ (0.89 \times (No.partners\ in\ past\ 3months = 1)) \\
&+ (0.70 \times (No.partners\ in\ past\ 3months = 2)) \\
&+ (1.93 \times (No.partners\ in\ past\ 3\ months = 3\ or\ more)) \\
&- (0.16 \times (condom\ use\ in\ last\ sex = yes)) \\
&+ (0.56 \times (Highrisk\ sex\ in\ past\ 3months = yes))
\end{aligned}
$$

on average at 12 months. These scores were then dichotomised into "risky" and "less-risky" sexual behaviour, using the mean baseline risk score value (-5.05) as the cut-off.

Among women who attended both baseline and 12 month visits, the proportion with risky behaviour at 12 months was significantly lower than at baseline (3.3% vs 12.8%, McNemar Chi-squared p<0.001).

### Baseline variables associated with risky sexual behaviour at 12 months

There was strong evidence that women with risky sexual behaviour at baseline had an increased odds of having risky behaviour at 12 months (OR 11.38, 95%CI = 5.86–22.10; p<0.001; Table 4).

After adjusting for risky behaviour at baseline, there was strong evidence that risky sexual behaviour at 12 months was associated with several baseline variables (Table 4). Single women had 2.5 times the odds of having risky behaviour at 12 months (adjusted (a)OR = 2.55, 95% CI = 0.81–8.02) as married women, as did women who were separated/divorced/widowed (aOR 2.47, 95%CI = 0.82–7.44). There was also strong evidence that women who reported two or more partners in the last 12 months, and those reporting more than five partners in their lifetime, had higher odds of having risky behaviour at 12 months than those reporting fewer partners (p<0.005). Transactional sex and harmful or hazardous drinking were associated with nearly four times the odds of having risky behaviour at 12 months (p<0.001).

On the other hand, being older than 16 years at sex debut was associated with a 75% reduction in the odds of risky behaviour at 12 months (aOR 0.25, 95%CI = 0.12–0.52; p<0.001). Similarly, being older than 25 years at enrolment was associated with a nearly 50% reduction in the odds of risky behaviour at 12 months (aOR 0.52, 95%CI = 0.37–1.02; p = 0.06).

## Discussion

This analysis showed that sexual risk behaviours among women known to be at higher HIV-risk enrolled in HIV prevention studies decreased over time. Specifically, the proportion of women reporting to have multiple partners, engage in transactional sex and in high-risk sex, consistently decreased at each follow-up visit (p for linear trend <0.001 for all). It was also observed that, women who had risky behaviours at enrolment were more likely to have more risky behaviour after one year follow-up. On the other hand, older age at sexual debut and at study enrolment were protective against adopting risky behaviours after 12 months irrespective of baseline behaviours.

This finding of a reduction in risk behaviours could imply that involvement in a study that provided regular HIV/STI testing, risk reduction counselling (every three months) and access to condoms and prevention information, may have altered participant's sexual risk behaviour. The HIV/STI testing and counselling alone may have led the participants to reflect about their risk behaviour as we observed that, at each subsequent study visit, there was a 13% increase in proportion of women who considered themselves to be at higher risk of HIV seroconversion (p for linear trend<0.001). It is likely that the regular behaviour assessment including questions on condom use and other sexual and drug related behaviours may have encouraged self-reflection or increased motivation for improved safe sex. This behavioural reactivity in the absence of a direct intervention has been documented in observational studies evaluating sexual risk, mental health and substance use among female sex workers (FSWs) [20, 21].

Women who had risky behaviour at enrolment were more likely to have risky behaviour after one year follow-up than those with less risky behaviour at baseline. Risk reduction counselling as noted above, was not always effective in this group of women. Similar findings have been reported in other HIV vaccine feasibility studies showing that individuals who engage in

**Table 4. Factors associated with risky sexual behaviour at 12 months, among 1180 women attending the baseline and 12 months visits.**

| Variable | n/N (row%) | Unadjusted OR (95%CI) | P-value | Adjusted OR [†] (95%CI) | P-value |
|---|---|---|---|---|---|
| **Baseline sexual behaviour risk score** | | | | | |
| Less-risky behaviours (≤ -5.05) | 16/1029(2) | 1 | <0.001 | N/A | N/A |
| Risky behaviours (>-5.05) | 23/151(15) | 11.38(5.86–22.10) | | | |
| **Age at enrolment (years)** | | | | | |
| <25 | 22/417(5) | 1 | 0.006 | 1 | 0.06 |
| ≥25 | 17/763(2 | 0.41(0.21–0.78) | | 0.52(0.37–1.02) | |
| **Town of residence** | | | | | |
| Moshi | 5/368(1) | 1 | 0.017 | 1 | 0.10 |
| Lake zone | 34/812(4) | 3.17(1.23–8.17) | | 2.28(0.86–6.01) | |
| **Marital status** | | | | | |
| Married | 4/325(1) | 1 | 0.007 | 1 | <0.001 |
| Single | 14/354(4) | 3.30(1.08–10.14) | | 2.55(0.81–8.02) | |
| Separated /divorced/widowed | 21/501(4) | 3.51(1.19–10.32) | | 2.47(0.82–7.44) | |
| **Educational Level**<br>None-Primary<br>Secondary-Tertiary | 35/984(4)<br>4/194(2) | 1<br>0.57(0.20–1.62) | 0.26 | 1<br>0.55(0.19–1.60) | 0.27 |
| **Age at sexual debut(years)** | | | | | |
| <16 | 24/308(8) | 1 | <0.001 | 1 | <0.001 |
| ≥16 | 13/785(2) | 0.20(0.10–0.40) | | 0.25(0.12–0.52) | |
| **Number of partners past 12m** | | | | | |
| 0–1 | 3/590(0.5) | 1 | <0.001 | 1 | 0.001 |
| 2+ | 35/527(7) | 13.92(4.26–45.53) | | 7.69(2.25–26.31) | |
| **Number of lifetime partners** | | | | | |
| 0–4 | 5/631(0.8) | 1 | <0.001 | 1 | 0.004 |
| ≥ 5/don't know | 34/547(6) | 8.30(3.22–21.37) | | 4.39(1.62–11.92) | |
| **Transactional sex past 3m** | | | | | |
| No | 12/875(1) | 1 | <0.001 | 1 | 0.001 |
| Yes | 26/299(9) | 6.85(3.41–13.76) | | 3.80(1.78–8.07) | |
| **AUDIT score** | | | | | |
| Non-drinker or low-risk | 21/1030(2) | 1 | <0.001 | 1 | 0.001 |
| Harmful or hazardous drinking | 18/144(13) | 6.86(3.56–13.23) | | 3.52(1.71–7.22) | |
| **HIV risk perception** | | | | | |
| Small/no risk/ doesn't know | 22/886(3) | 1 | 0.009 | 1 | 0.08 |
| Moderate/great risk | 17/290(6) | 2.45(1.28–4.67) | | 1.83(0.93–3.62) | |

Analysis restricted to participants who attend both baseline and 12 months visit (N = 1180).n = Number with outcome of interest (risky behaviour risks score at 12 months).

[†] Estimated OR adjusted only for baseline sexual behaviour risk score.

high risk behaviours prior to cohort enrolment are more likely to sustain high risk behaviours during cohort participation [13]. It is also possible that, after repeated HIV negative serological test results at follow-up visits, women who had more risky behaviours at enrolment continued to engage in these practices assuming that the behaviours were low-risk. Another possible explanation is that, since the interventions provided in the feasibility cohorts did not address the economic circumstance of the women, women engaging in transactional sex at baseline (which was associated with nearly four times the odds of having risky behaviour at 12 months) may have persisted in their risky behaviour for financial reasons. Some women in this cohort had day jobs but engaged in transactional sex to supplement their income. Recent studies

among women at high HIV risk such as those in this cohort have shown evidence that cash transfers and financial support significantly reduce risky behaviour and consequently HIV risk [22].

While there was low condom use with regular partners and a significant decrease in this over subsequent visits, condom use with non-regular partners did not change with time. Reduction in unprotected sex after one year of follow-up has been reported in other studies and in meta-analyses of HIV risk reduction interventions. However, these studies did not specify if these were observed in regular or non-regular partnerships which is an important distinction for women at high HIV risk such as those in this study [14] [23, 24]. Low condom use with regular partners among women at high HIV risk has been extensively reported in previous literature [8, 11, 25–27] [28]. In one such study by Lowndes et in Benin [29], it was shown that a greater percentage of FSW's regular partners were HIV positive compared to their non-regular partners (16.1% vs. 8.5%).Therefore, given the multiple and concurrent partnerships among women in these high risk populations, unprotected sex with regular partners may represent their greatest risk of HIV infection. It has also been reported that such women feel powerless in negotiating condom use with their regular partners for fear of losing their emotional and economic support[11, 26]. This underscores the need for alternative prevention strategies to replace or supplement condom use in this population for instance, pre-exposure prophylaxis (PrEP) in oral or topical microbicide form.

This study has several implications for the design of future studies which have HIV acquisition as an outcome, including HIV vaccine feasibility and intervention studies. First, since planning for HIV vaccine efficacy trials requires identification of populations with high HIV incidence, the identified factors observed to be associated with risky behaviour at 12 months may provide a guide in recruitment of the women with the highest risk of HIV acquisition. For instance, individuals who have a history of high-risk behaviour will be more likely to continue engaging in high-risk behaviours during the trial. Also, special efforts could be made to ensure optimal follow-up of these participants, who are also more likely to acquire HIV. These strategies could potentially reduce the required sample size and follow-up time needed in studies which have HIV incidence as an outcome. Secondly, understanding patterns of changes in behaviour and their associated factors underscores the need to tailor and provide intensified HIV risk reduction interventions. This study highlights the ethical paradigm to provide the best care to participants in vaccine trials as the findings indicate that, the current practices of risk reduction counselling were not always sufficient to change behaviours of those women with the highest HIV risk. There is therefore, a need to ensure that interventions such as PrEP are readily made available to complement behavioural interventions among participants who continue to engage in risky behaviours during follow-up.

A notable limitation of the study–as in most behavioural research–is that, the data are self-reported and therefore subject to social desirability and recall bias [30–32]. Women may have under-reported their number of partners or over-reported condom use so as to please the research staff. This may have resulted in differential misclassification of the changes in sexual behaviour as women with high risk behaviour at baseline would likely report a reduction in their risky behaviour at the 12-month visit. This may have consequently biased the observed OR and over-estimated the reduction in risky behaviour. The study made attempts to minimize reporting bias by making the questions neutral, assuring participants of confidentiality and having frequent follow-up visits as further detailed in the methodological paper [15]. Loss to follow-up bias is also of concern as there was some evidence of a differential loss to follow-up in a sub-analysis performed to characterize baseline characteristics of women who did not complete follow-up. Women who did not attend the 12-month visit (194 women) were younger, unmarried and arguably "at higher-risk". It is likely that reduction in sexual behaviours

was overestimated as a result of early loss to follow-up of higher-risk women because, the proportion of women with higher number of partners, those engaging in transactional sex and in high-risk sex was significantly higher among women dropping out than those remaining in the study. Lastly, due to the limited number of HIV seroconversions (44 events) the predicted sexual behaviour risk score was based on only three behaviour variables. The criteria used in building the Poisson model were subjective with stringent criteria used in selection of the covariates. This may weaken the external validity of the risk score when assessing risk behaviours in a dataset with more HIV seroconversions.

In spite of these limitations, the primary strength of the study was the prospective cohort design which enabled estimation of changes in behaviour in each subsequent visit. The inclusion of women from two cohorts conducted with comparable designs provided an adequate sample size powered for many components of this analysis. The predictive risk score included behavioural variables that are commonly collected in routine HIV services. The risk score developed may be specific to the particular study population and may not be generalizable to other contexts. Therefore, for application in a different setting, a risk score would need to be developed and carefully validated Lastly a good retention rate of 86% over 12 month follow-up was attained in our study, in spite of the high mobility of the study population.

## Conclusions

Taken together, our findings serve as an important reminder that, risk reduction counselling and access to HIV prevention interventions in cohorts of high-risk women are important and do result in some reduction in risk behaviour. Qualitative research is needed to better understand participants' perspectives on participating in trials and the extent to which increased risk behaviour may result from various aspects of trial participation. Furthermore, risk reduction counselling in itself is not enough, as a proportion of participants continued to engage in high risk sexual behaviours. This underscores the importance of biomedical HIV preventions interventions such as PrEP to most at risk individuals to supplement risk-reduction counselling and behavioural intervention so as to further minimise HIV seroconversion risk.

## Acknowledgments

The authors would like to thank all participants who agreed to take part in this study; research assistants, laboratory technicians and administrative staff for all their efforts in the implementation of both the microbial and vaccine feasibility studies.

## Author Contributions

**Conceptualization:** Diana Faini, Claudia Hanson, Kathy Baisley, Saidi Kapiga, Richard Hayes.

**Data curation:** Richard Hayes.

**Formal analysis:** Diana Faini, Kathy Baisley, Saidi Kapiga, Richard Hayes.

**Funding acquisition:** Saidi Kapiga, Richard Hayes.

**Investigation:** Saidi Kapiga, Richard Hayes.

**Methodology:** Diana Faini, Kathy Baisley, Saidi Kapiga, Richard Hayes.

**Project administration:** Saidi Kapiga.

**Supervision:** Claudia Hanson, Kathy Baisley, Saidi Kapiga, Richard Hayes.

**Validation:** Saidi Kapiga, Richard Hayes.

**Writing – original draft:** Diana Faini.

**Writing – review & editing:** Diana Faini, Claudia Hanson, Kathy Baisley, Saidi Kapiga, Richard Hayes.

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
