## [Decision Letter · Decision Letter 0]

3 Feb 2020

PONE-D-19-26501

Sexual behaviour, changes in sexual behaviour and associated factors among women at high risk of HIV participating in feasibility studies for prevention trials in Tanzania

PLOS ONE

Dear Dr Faini,

Thank you for submitting your manuscript to PLOS ONE. After careful consideration, we feel that it has merit but does not fully meet PLOS ONE’s publication criteria as it currently stands. Therefore, we invite you to submit a revised version of the manuscript that addresses the points raised during the review process.

The manuscript has been assessed by two reviewers, their comments are available below. The reviewers find the work of relevance but have raised some items that need attention in a revision. The reviewers request clarifications about the analyses undertaken, recommend some further analyses and request additional discussion of limitations of the study.

Could you please revise the manuscript to address the items raised.

We would appreciate receiving your revised manuscript by Mar 17 2020 11:59PM. Please include the following items when submitting your revised manuscript:

We look forward to receiving your revised manuscript.

Kind regards,

Iratxe Puebla

Deputy Editor-in-Chief, PLOS ONE

Journal Requirements:

2. Please refer to any post-hoc corrections to correct for multiple comparisons during your statistical analyses. if these were not performed please justify the reasons. Additionally, please include any participant exclusion criteria within the manuscript.

Reviewers' comments:

Reviewer's Responses to Questions

**Comments to the Author**

1. Is the manuscript technically sound, and do the data support the conclusions?

Reviewer #1: Yes

Reviewer #2: Partly

2. Has the statistical analysis been performed appropriately and rigorously? 

Reviewer #1: Yes

Reviewer #2: Yes

3. Have the authors made all data underlying the findings in their manuscript fully available?

Reviewer #1: Yes

Reviewer #2: Yes

4. Is the manuscript presented in an intelligible fashion and written in standard English?

Reviewer #1: Yes

Reviewer #2: No

5. Review Comments to the Author

Reviewer #1: It’s an interesting study that needs minor revision The authors used a random intercept logistic regression model to estimate odds ratios for the change in prevalence of certain risky behaviours. However, the justification for this multilevel modeling is not clearly stated. For example, is it to cater for variance between visits and correlation between individual repeated measurements? It would have been appropriate to specify the random intercept logistic regression model used for clarity.

The authors said “since few women were categorized as having risky sexual behavior at 12months, no attempt was made to build a full multivariable model to avoid problems with data sparsity” but it’s not clear how factors associated with risky behavior at 12 months were identified (table4 of results).

Lastly, self-reported changes in risky behavior is such a highly subjective matter prone to bias especially where the respondent is aware of the purpose of the study. This may have overestimated the effect

Reviewer #2: The grammar is largely correct ecxept in a few places where minor corrections are needed. The authors need to make some corrections to the few typos. Statistical analysis is clear except in Poisson regression.

6. PLOS authors have the option to publish the peer review history of their article (what does this mean?). If published, this will include your full peer review and any attached files.

Reviewer #1: No

Reviewer #2: No

---

## [Author Response · Author response to Decision Letter 0]

19 Mar 2020

Reviewer #1, Comment 1:

It’s an interesting study that needs minor revision The authors used a random intercept logistic regression model to estimate odds ratios for the change in prevalence of certain risky behaviours. However, the justification for this multilevel modelling is not clearly stated. For example, is it to cater for variance between visits and correlation between individual repeated measurements? It would have been appropriate to specify the random intercept logistic regression model used for clarity.

Response:

We thank the reviewer for this important question. Yes, as the reviewer points out, the justification for using Logistic regression with random effect intercepts was to account for correlations between repeated measurements at the multiple visits. A sentence has been added to the manuscript to clarify this as suggested by the reviewer . (Page 6, lines 132-133).

Reviewer #1,Comment 2: 

The authors said “since few women were categorized as having risky sexual behaviour at 12 months, no attempt was made to build a full multivariable model to avoid problems with data sparsity” but it’s not clear how factors associated with risky behaviour at 12 months were identified (table 4 of results).

Response:

We thank the reviewer for this comment. The Table four (4) of results presents the outcome of the Logistic regression model which examines each covariate with its association with the “having risky behaviour at 12 months” (outcome of interest). Following the univariate association, each covariate was adjusted for the effect of the “baseline behaviour”. While this is not a full multivariable model, each covariate was only adjusted for the effect of the baseline behaviour. Therefore, the presented adjusted odds ratios are “partially” adjusted as they do not take into account the effect of the other behaviours. (Page 7,line 162-166).

We have revised the manuscript in the section quoted by the reviewer so as to improve clarity of the analyses performed. A footnote in Table 4 explains that the estimated odds ratio are adjusted only for the baseline sexual behaviour risk score. 

Reviewer #1, Comment 3:

Lastly, self-reported changes in risky behaviour is such a highly subjective matter prone to bias especially where the respondent is aware of the purpose of the study. This may have overestimated the effect.

Response:

We agree with the reviewer that self-reported behaviours are prone to reporting bias i.e the social desirability bias. It is for this reasons that we have acknowledge and discussed this as a study limitation and how it may have overestimated the reported reduction of risky behaviour. 

In the discussion section, we have also highlighted the several attempts made to minimize this bias during study design and data collection. To underscore the effect of this bias on our study findings, we have reviewed and made necessary additions to this section in the manuscript. (Page 20, line 311-317).

Reviewer #2 comment 1: 

The grammar is largely correct except in a few places where minor corrections are needed. The authors need to make some corrections to the few typos. 

Response:

We thank the reviewer for pointing this. We have reviewed the entire manuscript and take note of the grammar and typographical errors. We have corrected the errors and proof-read the final version of the manuscript. 

Reviewer #2 comment 2: 

Statistical analysis is clear except in Poisson regression.

Response:

We take note of the reviewer’s concern on the clarity of the Poisson regression analysis performed. We have reviewed this section in detail and revised the paragraph to improve clarity. (Page 7, line 138-154).

---

## [Editor Report · Decision Letter 1]

1 Apr 2020

Sexual behaviour, changes in sexual behaviour and associated factors among women at high risk of HIV participating in feasibility studies for prevention trials in Tanzania

PONE-D-19-26501R1

Dear Dr. Faini,

We are pleased to inform you that your manuscript has been judged scientifically suitable for publication and will be formally accepted for publication once it complies with all outstanding technical requirements.

With kind regards,

Ethan Morgan

Academic Editor

PLOS ONE
---

## [Editor Report · Acceptance letter]

6 Apr 2020

PONE-D-19-26501R1 

Sexual behaviour, changes in sexual behaviour and associated factors among women at high risk of HIV participating in feasibility studies for prevention trials in Tanzania 

Dear Dr. Faini:

I am pleased to inform you that your manuscript has been deemed suitable for publication in PLOS ONE. Congratulations! Your manuscript is now with our production department. 

With kind regards,

on behalf of

Dr. Ethan Morgan 

Academic Editor

PLOS ONE